# Exploring causal effects of smoking and alcohol related lifestyle factors on self-report tiredness: A Mendelian randomization study

**Heshan Li[1◉], Junru Zhao[2◉], Jing Liang[3◉], Xiaoyu Song[1]***

**1** School of Life Sciences and Technology, Harbin Institute of Technology, Harbin, China, **2** School of Mathematical Sciences, Harbin Engineering University, Harbin, China, **3** Harbin Huaqiang Power Automation Engineering Company Limited, Harbin, China

◉ These authors contributed equally to this work.
* songxyhit@hit.edu.cn

**Data Availability Statement:** All data files are available from the IEU open GWAS database. https://gwas.mrcieu.ac.uk/.

## Abstract

Self-reported tiredness or low energy, often referred to as fatigue, has been linked to lifestyle factors, although data from randomized–controlled trials are lacking. We investigate whether modifiable lifestyle factors including smoking and alcohol intake related exposures (SAIEs) are causal factors for fatigue using Mendelian randomization (MR). A two-sample MR study was performed by using genome-wide association summary results from UK Biobank (UKBB), and each of the sample size is more than 100,000. We used the inverse variance weighted method, and sensitivity analyses (MR Egger, weighted median, penalized median estimators, and multivariable MR) to account for pleiotropy. The two-sample MR analyses showed inverse causal effect of never-smoking status and positive effect of current smoking status on the risk of fatigue. Similarly, genetically predicted alcoholic intake was positively associated with fatigue. The results were consistent across the different MR methods. Our Mendelian randomization analyses do support that the cessation of smoking and alcohol can decrease the risk of fatigue, and limit alcohol intake frequency can also reduce the risk.

## Introduction

Self-reported tiredness and low energy are often called fatigue which is widespread in the population [1–5]. It is a common presentation in primary care [6–9]. While many lifestyle factors have been associated with fatigue, the causality of the association of lifestyle factors with fatigue remains unestablished. The first challenge is that demonstrating the causality with observational epidemiological studies is infeasible due to biases such as confounding and reverse causation. One way to assess causality is with Mendelian randomization (MR), which uses genetic variants as instrumental variables in an approach analogous to a randomized controlled trial, to assess whether risk factors have causal associations with an outcome of interest [10–12].

Another challenge is that fatigue is both heterogeneous (occurring in different conditions) and multifactorial, and it is commonly unexplained by underlying disease [7, 13]. It is the complexity of fatigue that makes it infeasible to discern its causal link with lifestyle factors via

**Funding:** The study was supported by the funds of Ningbo NMR company limited (MH2016017). website: www.xmc.cn.

**Competing interests:** The authors have declared that no competing interests exist.

observational studies. Most epidemiological data examining risk factor modification have studied the relationships between modifiable risk factors and fatigue in a specific sample and under a certain condition.

We performed a two-sample Mendelian randomization analysis—including inverse variance weighted (IVW), MR-Egger regression, weighted median estimator, and penalized weighted median estimator—to investigate the etiological role of the two most important modifiable lifestyle factors (smoking and alcohol intake related exposures) on fatigue with unlimited samples. Multivariable MR analysis was performed as additional sensitivity analyses since psychiatric disorders frequently occur comorbidly and share similar phenomenological features of fatigue. This multivariable approach takes psychiatric disorders (bipolar disorder, schizophrenia and depression) into account to explore whether there are independent direct casual effects of SAIEs on fatigue.

## Methods

We undertook Mendelian randomization analyses to estimate effects of SAIEs on fatigue in a two-sample Mendelian randomization framework. Mendelian randomization is a method of using genetic variants reliably related to a modifiable exposure to obtain evidence regarding the causal influence of the exposure on disease in observational studies. This is achieved through the properties of genetic variants with nonsusceptibility to reverse causation and confounding, which otherwise bedevil epidemiological studies [14]. Two-sample MR exploits the fact that it is not necessary to obtain the effect of the instrumental variable-risk factor association and the instrumental variable-outcome association from the same sample of participants. Two-sample MR allows the use of freely accessible summary association results from GWAS to estimate causal effects of modifiable exposures on outcomes. The main advantage of two-sample MR is the increased statistical power [15].

The Mendelian randomization approach was based on the following assumptions, which have been widely described in recent studies [16–18]:

1. The genetic variants used as instrumental variables (IVs) are predictive of the exposure.

2. IVs are independent of any confounders of the exposure-outcome relationship.

3. IVs are associated with outcomes only through the clinical risk factors, that is, a lack of pleiotropy (Fig 1).

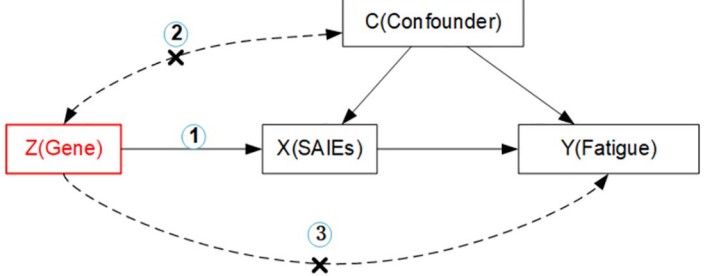

**Fig 1. A directed acyclic graph illustrating core instruments variable assumptions of the Mendelian randomization approach.** IV assumption 1: IVs are strongly correlated with exposures. IV assumption 2: IVs are independent of outcomes (i.e., IVs can only affect outcomes through exposures). IV assumption 3: IVs are not related to confounding factors. IV: instrument variable; SAIEs: smoking and alcohol intake related exposures.

These assumptions imply that there is only one causal pathway from the genetic variant to the outcome via the risk factor, and no other causal pathway either directly or indirectly to the outcome [16]. A diagram corresponding to these assumptions is presented in Fig 1.

Publicly available summary results of SAIE—effect estimates and their standard errors for each single-nucleotide polymorphism (SNP) 's effect on exposures and fatigue—were used for the main Mendelian randomization analysis and therefore no additional ethics approval was required.

### Two-sample MR analyses

**Data sources and processing.**   Genome-wide significant ($P < 5 \times 10^{-8}$) genetic variants of predicted SAIE and psychiatric disorders (bipolar disorder, schizophrenia and depression) were extracted to satisfy IV assumption 1, and the insignificant genetic variants of self-reported fatigue were extracted to satisfy IV assumption 3. Consequently, their intersection was analyzed by MR analyses.

Genetic variants should be further processed into smaller datasets that meet IV assumptions. First of all, we removed genetic variants with linkage disequilibrium (LD). LD induces correlation between two genetic variants, and destroys the randomness of genetic variants [19, 20]. After SNPs with potential LD were removed, the parts with longer physical distance (more than 10,000kb) and less possibility of LD ($R^2 < 0.001$) were retained. Besides, genetic variants of palindromic and incompatible alleles should be removed when we harmonized the exposure and outcome datasets.

*Smoking related exposures.* 70 SNPs that were significantly associated with never-smoking status were extracted from summary results data, which belongs to a GWAS of UKBB (N = 359,706 individuals of European ancestry). We also extracted 13 SNPs significantly associated with current-smoking status from summary results data, which belongs to a GWAS of UKBB (N = 336,024 individuals of European ancestry). The SNPs significantly associated with smoking status are listed in Table A in S1 Table.

*Alcohol intake related exposures.* Only one SNP that was significantly associated with never-drinking alcohol status was extracted from summary results, which belongs to a GWAS of UKBB (N = 336,965 individuals of European ancestry). We extracted three SNPs significantly associated with current-drinking alcohol status, which belong to summary results from a GWAS of UKBB (N = 360,726 individuals of European ancestry). Besides this, we extracted 90 SNPs significantly associated with alcohol intake frequency (AIF) from summary results, which belong to a GWAS of UKBB (N = 462,346 individuals of European ancestry). The SNPs significantly associated with alcohol intake are listed in Table C in S1 Table.

*Self-reported tiredness.* We obtained summary results statistics from the GWAS of UKBB. The GWAS of self-reported tiredness included 449,019 individuals, who reported their own tiredness or lethargy in last weeks. We extracted 8922284 SNPs from the GWAS summary results ($P < 0.05$).

*Psychiatric disorders.* IEU open GWAS provided the summary staistics of European ancestry for bipolar disorder (IEU identifier: ieu-b-42, N = 77,096 individuals) and schizophrenia (IEU identifier: ieu-b-41, N = 51,710 individuals). UKBB provided the summary statistics of European ancestry for depression (UKBB identifier: ukb-d-F5_DEPRESSIO, N = 361,194 individuals). We extracted the SNPs significantly associated with bipolar disorder, schizophrenia and depression, corresponding to 12625, 240 and 5 respectively ($P < 5 \times 10^{-8}$).

**Statistical analysis.**   To investigate the causal relationship between SAIE and fatigue, we initially performed two-sample inverse variance weighted MR analyses by the SNPs extracted from GWAS summaries [21]. The magnitude of the causal effect ($\hat{\beta}_{IVW}$) was estimated as the

average of the SNP-outcome effect ($\hat{\beta}_{ZY}$) divided by the SNP-exposure effect ($\hat{\beta}_{ZX}$). The regression slope (ratio) in IVW analysis is forced through a zero intercept. Analyses were performed with the Two-Sample MR package [22].

To avoid the violation of the IV assumptions 2 and 3, we assessed instrument strength for the standard IVW MR analysis via calculating the approximate F statistics for each of exposures that will be used in our two-sample MR studies [i. e. $F \approx (\beta/SE)^2$ shown in Table A and C in S1 Table, SE: standard error.].

To test whether all the IVs satisfy the IV assumptions 2 and 3, we performed a test of heterogeneity of causal effect estimates across each of the SNPs via Cochran's Q value quantifying heterogeneity and detecting outliers. If an individual SNP's Q contribution is extremely large (i.e., above the 5% threshold of 3.84 or instead of a Bonferroni-corrected threshold), it might imply heterogeneity, including horizontal pleiotropy [23, 24]. Then we used MR Egger regression to further investigate the possibility of directional pleiotropy in our data (i.e. where some SNPs influence the outcome via additional paths other than via the exposure), and further verified such pleiotropy with funnel plots, which plot instrument strength against the causal estimates for all the IVs. Asymmetry in funnel plots suggests that pleiotropic effects are not balanced and may indicate directional pleiotropy.

## Sensitivity analyses

As the resulting IVW analysis suffers from bias and invalid IVs [25], we performed sensitivity analyses via three additional MR models, i.e., MR Egger regression, the weighted median estimator, penalized weighted median estimator, to obtain more robust estimates [26]. All the following sensitivity analyses were subsequently performed upon exclusion and inclusion of outliers respectively, which were identified through Cochran's Q statistics.

In the case of MR Egger regression analysis, we also assessed instrument strength using an $I_{GX}^2$ statistic because F statistics from the individual marks are not sufficient indicators of instrument strength [27]. Bowden showed that an $I_{GX}^2$ statistic can be used to quantify the expected dilution of MR-Egger regression estimates [27]. He showed that a high value of $I_{GX}^2$ (i.e. close to one) would indicate no dilution. On the contrary, if $I_{GX}^2$ is less than 0.9, inference from MR Egger should be interpreted with caution, and some alternative sensitivity analyses should be considered [28, 29].

In contrast to the weighted mean used in the IVW analysis, the weighted median estimator uses the weighted median of the ratio (i.e., the ratio of $\hat{\beta}_{ZY}$ and $\hat{\beta}_{ZX}$). In the weighted regression model, the penalized weighted median approach downweighs (or penalizes) the contribution of some candidate variants with heterogeneous ratio estimates, and may have better finite sample properties than weighted median, particularly if there is directional pleiotropy [26].

If the four MR models (IVW, MR Egger regression, weighted median estimator, penalized weighted median), which make different assumptions regarding instrument validity, produce similar estimates of causal effects, then we will be more confident in the robustness of our findings.

## Mutivariable Mendelian Randomization analysis

IVW multivariable MR was performed as additional sensitivity analyses to test the robustness of the findings from univariate MR. It identified whether there are independent direct casual effects of smoking and alcohol intake on fatigue, when conditioning on the psychiatric disorders as possible mediators for fatigue.

The multivariable IVW regression was performed by regressing $\hat{\beta}_{ZY}$ on $\hat{\beta}_{ZX}$ (which was calculated by using all the SNPs at once with the causal effect for each exposure estimated from regression) in a single regression model. A critical assumption, additional to the univariate assumptions, for multivariable MR is that the relationship between genetic instruments and the outcome is only mediated by the exposures in the analysis [30].

We used the Two-Sample MR package (version 0.5.6) in R (version 4.1.3) to perform IVW, MR Egger regression, weighted median, IVW MR and penalized median estimator. We used the codes provided on https://mrcieu.github.io/TwoSampleMR/ to run analyses.

## Results

### Smoking status

In this section, we analysed two types of smoking status (never-smoking status vs current-smoking status) to investigate the causal effects of smoking exposures on fatigue outcome.

**Never-smoking status.** The two-sample MR IVW analysis showed evidence for a negative causal effect of never-smoking status on fatigue. This was supported by median-based model and penalized MR model both before and after outliers extraction (Figs 2 and 3). Multivariable IV analysis provided an additional evidence of a causal effect of never-smoking status on fatigue (Fig 4).

There was evidence of heterogeneity in the causal effect estimates of never-smoking status on fatigue across the individual SNPs (Q = 172.4, P = $8.21 \times 10^{-11}$). On the one hand, the estimate of the intercept of MR Egger regression showed no evidence of directional horizontal pleiotropy for never-smoking status exposure (Egger intercept was -0.0012 and its p-value was 0.53). But MR Egger result from never-smoking status analysis would be unreliable due to the low $I^2_{GX}$ statistic of 0.67 which was below the suggested cut-off of 0.9, suggesting that the MR Egger results could be influenced by measurement error or weak instruments bias. On the other hand, there was an abundance of asymmetry of the funnel plots suggesting there was directional horizontal pleiotropy among our data (Fig A in S1 File).

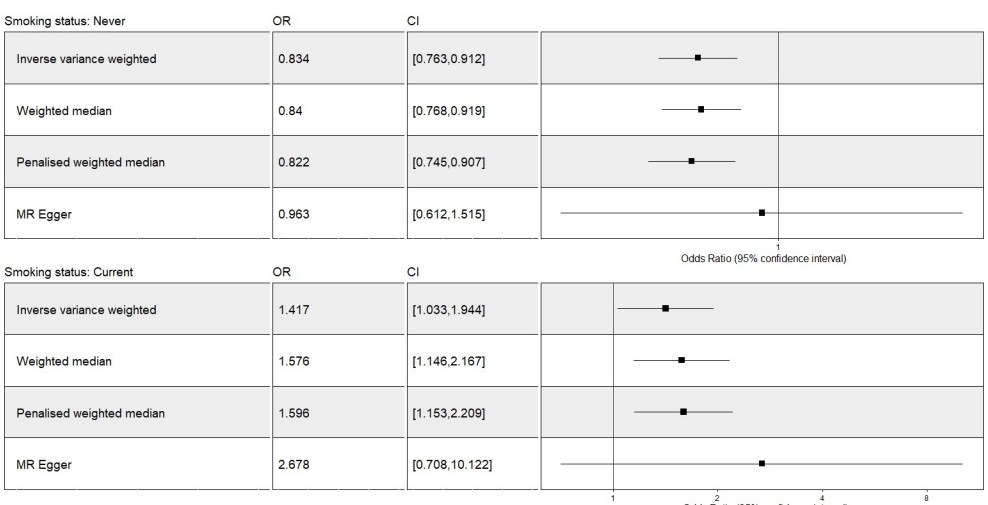

**Fig 2. The Mendelian Randomization analysis of smoking status (never vs current) on fatigue including outliers.** OR = odds ratio per unit decreases in smoking status (never vs current). Forest plot comparing results from inverse variance weighted, weighted median, penalized median and MR Egger methods. CI: confidence interval; MR: Mendelian Randomization.

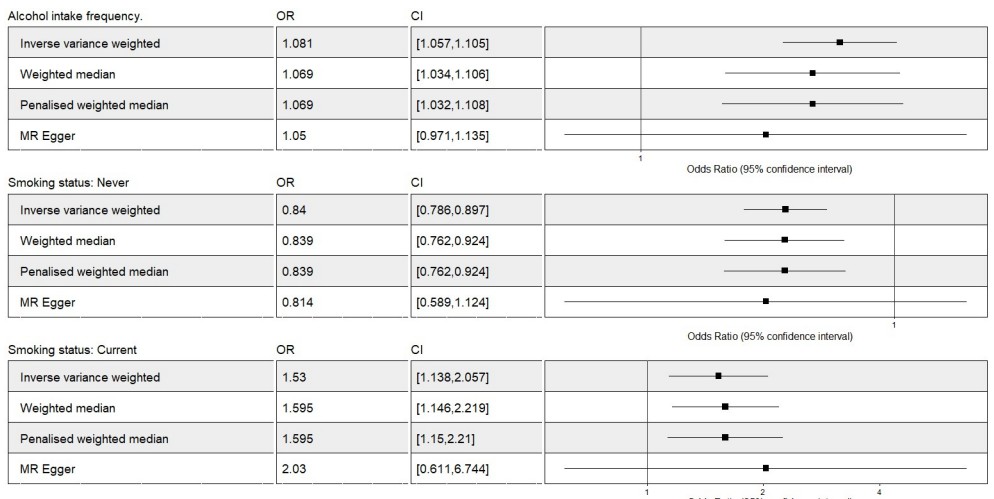

**Fig 3. The Mendelian Randomization analysis of SST and AIF on fatigue after outliers extraction.** OR = odds ratio per unit decreases in SST and AIF. Forest plot comparing results from inverse variance weighted, weighted median, penalized median and MR Egger methods. CI: confidence interval; SST: smoking status: never vs current; AIF: alcohol intake frequency; MR: Mendelian Randomization.

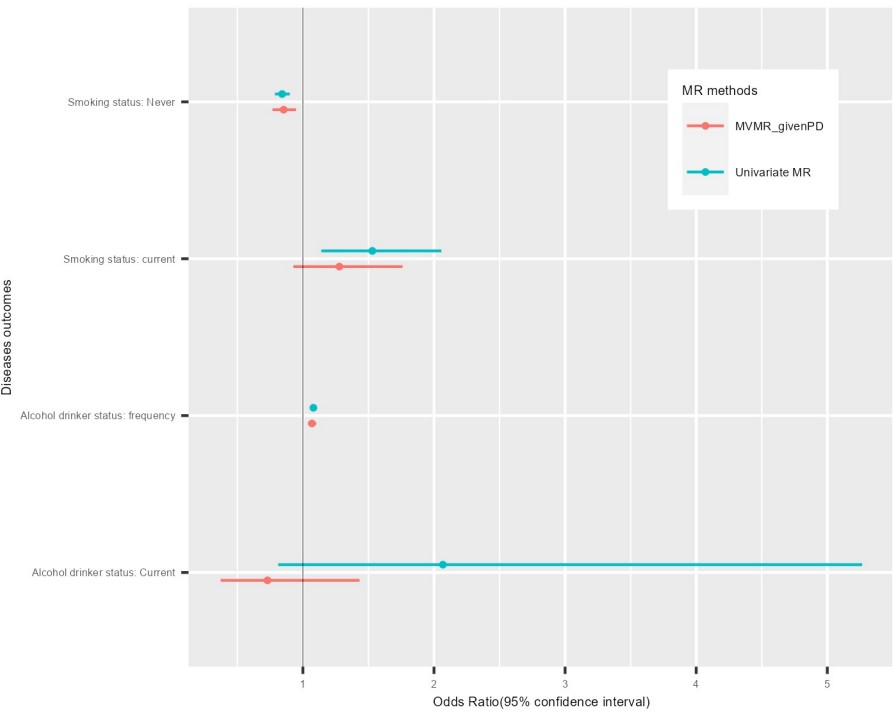

**Fig 4. The causal effects of SAIE on self-reported tiredness without (in blue) and with (in orange) adjusting for psychiatric disorders.** Dots and error bars show the odds ratio for estimated causal effect sizes and its 95% CI. SAIE: smoking and alcohol intake related exposures; CI: confidence interval; Mendelian randomization: MR; MVMR: multivariable Mendelian Randomization; PD: psychiatric disorders.

**Current smoking status.** The two-sample MR model showed evidence for a positive casual effect of current smoking status on fatigue both before and after the outliers extraction (Figs 2 and 3). This was also supported by median-based and penalized MR models. This estimated positive causal effect was not supported by IVW multivariable MR analysis (Fig 4). The discrepancy with our results—suggesting mediation or confounding by psychiatric disorders —highlights the importance of the inclusion for all the SNPs significantly associated with substance use and psychiatric disorders in multivariable MR analysis, to prevent loss of precision and potentially even biased estimates.

Our analysis suggested no significant evidence of horizontal pleiotropy (as indicated by MR-Egger regression intercept was -0.0035, with a *P* value larger than 0.05 shown in Table J in S1 Table). But the low $I^2_{GX}$ statistic of 0.86, below the suggested cut-off of 0.9, indicates that the MR Egger results would be unreliable. There was evidence of heterogeneity in the causal effect estimates of current-smoking status on fatigue across the individual SNPs (Q = 29.9, P = 0.0029). There was strong evidence of directionally horizontal pleiotropy because of an abundance of asymmetry of the funnel plots (Fig B in S1 File).

## Alcohol intake-related exposures

**Alcohol intake status.** The two-sample MR IVW analysis showed strong evidence for a negative causal effect of never alcohol intake status on fatigue while there was no evidence for causal effect of current alcohol intake status on fatigue (Fig 5). Multivariable IV analysis provided additional support for no causal effect of current alcohol intake status on fatigue (Fig 4).

**Alcohol intake frequency.** The two-sample MR IVW analysis showed evidence for a positive causal effect of AIF on fatigue. This was supported by median-based and penalized MR models (Figs 3 and 4). Multivariable IV analysis provided an additional evidence of a causal effect of AIF on fatigue (Fig 4).

There was evidence of heterogeneity in the causal effect estimates of AIF on fatigue across the individual SNPs (Q = 232.6, P = 1.15 $\times 10^{-15}$). The estimate of the intercept of MR Egger

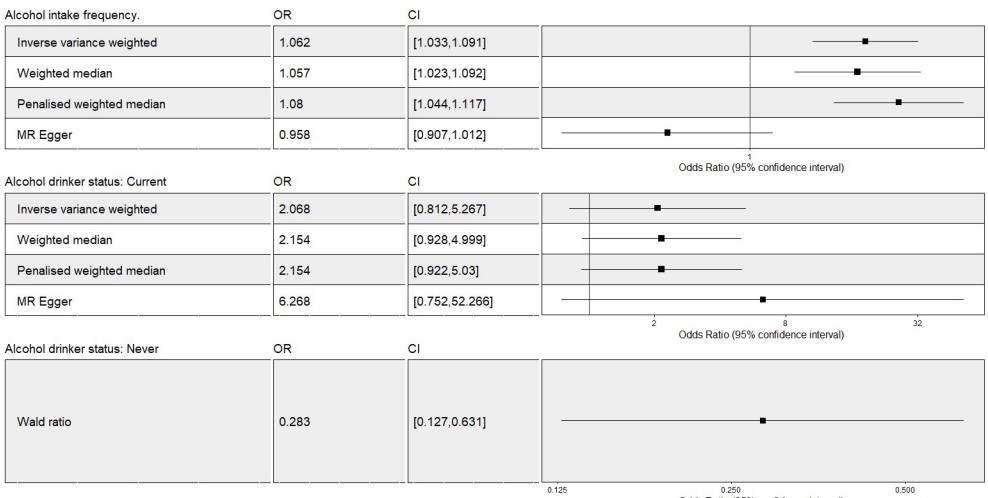

**Fig 5. The Mendelian Randomization analysis of alcohol intake-related exposures on fatigue including outliers.**
OR = odds ratio per unit decreases in alcohol intake related phenotypes. Forest plot comparing results from inverse variance weighted, weighted median, penalized median and MR Egger methods. There is only one genetic variant satisfied with the MR assumptions for the "never drinking" outcome. Therefore we use the Wald ratio as causal estimate here. CI: confidence interval; MR: Mendelian randomization.

regression showed strong evidence of directionally horizontal pleiotropy for AIF exposure (Egger intercept was 0.003 and its p-value was $8.43 \times 10^{-15}$). There was an abundant asymmetry of the funnel plots (Fig C in S1 File), suggesting there was directionally horizontal pleiotropy among our data.

## Discussions

### Smoking status

We found evidence of negative causal effect of never-smoking status on fatigue. The causal effect size estimates were consistent across the different sensitivity methods applied, and did not depend on whether or not there was direct pleiotropy in the analyses. Sensitivity analysis was performed with MR Egger regression, the weighted median estimator, and penalized median estimator. Additional MVMR analysis was also performed to investigate whether there is a direct casual effect of smoking status on fatigue. These analyses were consistent with the results from the IVW analyses. Thus, we generated that smoking is a risk factor to fatigue. We also found the positive causal effect of current-smoking status on fatigue in univariate MR analysis. But there is a discrepancy between the results from univariate MR and MVMR analyses. Such discrepancy suggests that the causal estimate of current-smoking status on fatigue is mediated or confounded by psychiatric disorders. This is supported by the observational evidence, the high rates of smoking in patients exist in mental illnesses [31]. Besides this, the association of smoking with some kind of psychiatric disorders might be bidirectional causality [32]. Thus psychiatric disorders can serve as mediators or confounders for current-smoking status exposure.

In former observational studies, smoking status has been associated with self-reported tiredness. Subjects who were heavy cigarette smokers were prone to fatigue. But this effect is only specific to the limited age range of participants rather than a broad population [33–35]. The effect estimates in observational study and our MR study are similar, providing additional confidence in our results. Our MR analysis goes one step further than observational studies. The former observational studies focused on healthy young males; our MR studies extended the sample to a wider range without age and gender limits.

Potential complication in interpreting the results of these sorts of MR analyses is the extent to which one can be sure that the causal estimates reflect the effect of smoking on energy levels. Low energy level refers to fatigue. Nicotine exposure from cigarette smoking can cause a negative energy state which is characterized by reducing energy intake and increasing energy expenditure. Energy intake and expenditure depend on brain feeding regulation [36, 37]. Nicotine can influence caloric consumption and energy expenditure by promoting the release of norepinephrine, serotonin, and other factors. These factors influence the brain to alter appetite and metabolic rate [37–40].

It is also possible that the causal estimates could reflect the effect of nicotine exposure on the increase of myocardial contractility and vasoconstriction, which results in increasing myocardial work and oxygen requirement and reducing coronary and cerebral blood flow [41]. Poor blood flow causes low energy level and then fatigue. Also, the heart must pump harder when circulation is poor, which can lead to further fatigue. Our study has proved that never-smoking reduces the risk of fatigue and current-smoking status exposure show no direct casual evidence on fatigue. Different smoking status will produce different gene expression patterns [42]. Non-smokers are not prone to fatigue.

### Alcohol intake-related exposures

**Alcohol intake status.** Evidence is lacking for causal effect of current-drinking status on fatigue. In contrast, we found some evidence of an inverse effect of never-drinking status on

fatigue. Such inconsistence of causal effect detection for alcohol intake may result from small number of SNPs associated with alcohol drinker statuses, which limits capacity to test causal effect [43].

**Alcohol intake frequency.** Evidence is enough for a causal relationship between AIF and fatigue. We found there is a statistically significant causal effect of AIF on fatigue. The causal effect size estimates of MR Egger regression were inconsistent across other different sensitivity methods applied. The low $I^2_{GX}$ for AIF partly indicated that sensitivity analysis of the resulting MR Egger regression was unreliable. The analyses from alternative pleiotropy-robust estimation strategies (including the weighted median estimator and penalized weighted model), complementary to MR Egger regression [44], were consistent with the results from the IVW analyses. The causal effect size estimate did not depend on whether or not there is direct pleiotropy in the analyses. Additional MVMR analyses further consolidated the finding of univariate MR.

Results from this study are consistent with previous conventional epidemiological studies which have associated alcohol intake with an increased risk of fatigue. But little evidence shows these apparently association may be causal [45]. Also, the subjects were drawn from a special fatigue clinic, so results cannot be generalised to other settings [45, 46]. Our MR study has extended this associate relationship to a causal one for a broad population without age and gender limits.

A significant causal relationship between alcohol intake and fatigue would reflect the detrimental effect level of alcohol on human physiology. Alcohol use actually inhibits the absorption and usage of vital nutrients such as vitamin B1, vitamin B12, folic acid, and zinc. Zinc is also essential to energy metabolic processes [47, 48]. Drinking alcohol constricts aerobic metabolism and decreases in hepatic ATP synthesis [49]. Thus the physical responses to alcohol in the body can lead to a feeling of fatigue and weakness.

## Strengths and limitations

One of the strengths of our MR study was the ability to select a large-scale SAIE GWAS data set as shown in Table J in S1 Table and fatigue (N = 449,019 individuals reporting their tiredness) in two-sample MR approach to estimate the effect of smoking status on fatigue, which helps overcome power limitations of MR. Second, participants in all the GWAS datasets are of European decent, which may reduce the influence on potential association caused by population stratification. The third strength is that we performed a series of sensitivity analyses to explore potential bias due to horizontal pleiotropy.

Due to the lack of relevant data, an important limitation with this study is that confounders associated with exposures and horizontal pleiotropy in our MR study were not fully explored. Besides, the small number of genetic instruments for some traits may have introduced weak instrument bias. Another limitation is that there is possible overlapping samples from the same GWAS dataset in two-sample MR. This will lead to the effects of over-fitting and weak instrument bias similar to those seen in one-sample MR. Our findings support several potential recommendations. In spite of the complexity of fatigue, individuals can still prevent it by changing lifestyles to reduce fatigue risks, such as smoking and alcohol intake cession.

## Conclusion

We found evidence for a causal effect of heavier alcohol intake on increasing the risk of fatigue. In addition, evidence is sufficient for a negative causal relationship of non-smoking on fatigue. The resulting MR analyses are consistent with previous observational studies with their relatively small sample size and the limited age range of participants.

## Supporting information

**S1 Table. Supplementary tables.** Table A: Effect of SNPs on alcohol intake frequency and alcohol drinker status (never vs current). The statistics are derived from linear regression. The GWAS summary data in the table is from UK Biobank. The GWAS identifiers of alcohol intake frequency and alcohol drinker status (never vs current) are corresponding to ukb-b-5779, ukb-a-226 and ukb-d-20117_2 respectively. 1The effect is the mean effect of the increaser allele estimated on a quantile normalized scale. The outlying genetic instruments which had large contribution to Cochran 's Q statistics are shown in bold. EAF: effect allele frequency; SNP: single nucleotide polymorphism; SE: Standard Error; GWAS: Genome-wide association study. Table B: The effect of the alcohol intake frequency and alcohol drinker status (never vs current) SNPs on self-reported tiredness. The statistics are derived from linear regression. The GWAS summary data in the table is from UK Biobank. The GWAS identifiers of alcohol intake frequency and alcohol drinker status (never vs current) are corresponding to ukb-b-5779, ukb-a-226 and ukb-d-20117_2 respectively. 1The effect is the mean effect of the increaser allele estimated on a quantile normalized scale. EAF: effect allele frequency; SNP: single nucle-otide polymorphism; SE: Standard Error; GWAS: Genome-wide association study. Table C: Effect of SNPs on smoking status (never vs current). The statistic are derived from linear regression. The GWAS summary data in the table is from UK Biobank. The GWAS identifiers of smoking status (never vs current) are corresponding to ukb-d-20116_0 and ukb-a-225 respectively. 1The effect is the mean effect of the increaser allele estimated on a quantile nor-malized scale. The outlying genetic instruments which had large contribution to Cochran 's Q statistics are shown in bold. EAF: effect allele frequency; SNP: single nucleotide polymorphism; SE: Standard Error; GWAS: Genome-wide association study. Table D: The effect of the smok-ing status (never vs current) SNPs on self-reported tiredness. 1 The effect is the mean effect of the increaser allele estimated on a quantile normalized scale. EAF: effect allele frequency; SE: Standard Error; SNP: single nucleotide polymorphism; GWAS: Genome-wide association study. Table E: Mendelian Randomization estimates of the causal effect of alcohol intake fre-quency on self-reported tiredness. Causal effects are estimated using four MR models: inverse variance weighted (IVW), weighted median, penalized median and MR Egger regression. Causal effect estimates are the difference in mean self-reported tiredness (in standard devia-tion; SD) per 1SD higher alcohol intake frequency. Significant results from Mendelian Ran-domization analysis are shown in bold. SE: Standard Error; IVs: Instrumental variables. Table F: Mendelian Randomization estimates of the causal effect of alcohol status (current) on self-reported tiredness. Causal effects are estimated using four MR models: inverse variance weighted (IVW), weighted median, penalized median and MR Egger regression. Causal effect estimates are the difference in mean self-reported tiredness (in standard deviation; SD) per 1SD higher alcohol status (current). Table G: Mendelian Randomization estimates of the causal effect of smoking status (never) on self-reported tiredness. Causal effects are estimated using four MR models: inverse variance weighted (IVW), weighted median, penalized median and MR Egger regression. Causal effect estimates are the difference in mean self-reported tiredness (in standard deviation; SD) per 1SD higher alcohol status (never). Significant results from Mendelian Randomization analysis are shown in bold. SE: Standard Error; IVs: Instru-mental variables. Table H: Mendelian Randomization estimates of the causal effect of smoking status (current) on self-reported tiredness. Causal effects are estimated using four MR models: inverse variance weighted (IVW), weighted median, penalized median and MR Egger regres-sion. Causal effect estimates are the difference in mean self-reported tiredness (in standard deviation; SD) per 1SD higher alcohol status (current). Table I: Results of heterogeneity tests. Significant results from the heterogeneity tests are shown in bold. Table J: Results of test for

directional horizontal pleiotropy. Significant results from the directional horizontal pleiotropy tests are shown in bold. SE: Standard Error.
(XLSX)

**S1 File. Supplementary results.** Fig A. Funnel plots for the effect of never smoking status on risk of self-reported fatigue for each single-nucleotide polymorphism (SNP), the resulting Mendelian randomization (MR) estimate is plotted against the inverse of the standard error of the MR estimate. Symmetry noted in this plot provides evidence against the presence of directional horizontal pleiotropy. The inverse-variance weighted and MR Egger causal estimates are represented by a red and blue line respectively. Fig B. Funnel plots for the effect of current smoking status on risk of self-reported fatigue for each single-nucleotide polymorphism (SNP), the resulting Mendelian randomization (MR) estimate is plotted against the inverse of the standard error of the MR estimate. Symmetry noted in this plot provides evidence against the presence of directional horizontal pleiotropy. The inverse-variance weighted and MR Egger causal estimates are represented by a red and blue line respectively. Fig C. Funnel plots for the effect of alcohol intake frequency on risk of self-reported fatigue for each single-nucleotide polymorphism (SNP), the resulting Mendelian randomization (MR) estimate is plotted against the inverse of the standard error of the MR estimate. Symmetry noted in this plot provides evidence against the presence of directional horizontal pleiotropy. The inverse-variance weighted and MR Egger causal estimates are represented by a red and blue line respectively.
(ZIP)

## Acknowledgments

This research has been conducted using the UK Biobank Resource and IEU open GWAS summary data. We would like to thank Ph.D. candidate Tao Wang from Peking University for help with statistical analyses.

## Author Contributions

**Conceptualization:** Xiaoyu Song.

**Data curation:** Heshan Li.

**Funding acquisition:** Xiaoyu Song.

**Investigation:** Heshan Li, Jing Liang.

**Methodology:** Junru Zhao, Jing Liang.

**Project administration:** Xiaoyu Song.

**Resources:** Heshan Li.

**Software:** Junru Zhao.

**Supervision:** Xiaoyu Song.

**Validation:** Jing Liang.

**Visualization:** Jing Liang.

**Writing – original draft:** Heshan Li, Jing Liang.

**Writing – review & editing:** Heshan Li, Jing Liang.

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
