## [Decision Letter · Decision Letter 0]

17 Jan 2023

PONE-D-22-28356

Exploring causal effects of smoking and alcohol related lifestyle factors on self-report tiredness: a Mendelian randomization study

PLOS ONE

Dear Dr. Song,

Thank you for submitting your manuscript to PLOS ONE. After careful consideration, we feel that it has merit but does not fully meet PLOS ONE’s publication criteria as it currently stands. Therefore, we invite you to submit a revised version of the manuscript that addresses the points raised during the review process.

The revised version should address all comments.

We look forward to receiving your revised manuscript.

Kind regards,

Petri Böckerman

Academic Editor

PLOS ONE

Journal Requirements:

Additional Editor Comments (if provided):

The revised version should address all comments. You may also note that the estimated effects may be sensitive to the use of different genetic scores (see https://doi.org/10.1002/hec.3828).

Reviewers' comments:

Reviewer's Responses to Questions

**Comments to the Author**

1. Is the manuscript technically sound, and do the data support the conclusions?

Reviewer #1: Yes

Reviewer #2: Yes

2. Has the statistical analysis been performed appropriately and rigorously? 

Reviewer #1: Yes

Reviewer #2: Yes

3. Have the authors made all data underlying the findings in their manuscript fully available?

Reviewer #1: Yes

Reviewer #2: Yes

4. Is the manuscript presented in an intelligible fashion and written in standard English?

Reviewer #1: Yes

Reviewer #2: Yes

5. Review Comments to the Author

Reviewer #1: Referee report on "Exploring causal effects of smoking and alcohol related lifestyle factors on self-report tiredness: a Mendelian randomization study" (PONE-D-22-28356).

The paper explores whether smoking and alcohol use cause fatigue using the MR method. A strength of the paper is that it uses several different approaches to study the topic. However, in my opinion, the methods should be more clearly explained, and previous evidence on the topic should be reviewed more carefully. I will elaborate below.

1. Plos One is an interdisciplinary journal and thus not all readers might be familiar with the MR method. Therefore, the idea of the two-sample MR and MR, in general, should be explained.

2. In the two-sample MR, the evidence on the associations of genetic variants with the risk factor and the outcome should come from nonoverlapping data sources. It seems that in this study, both associations come from the same sample (UKBB). Don’t the samples overlap? If so, what are the consequences?

3. There are very few citations to the previous literature regarding the relationship between smoking, alcohol intake and fatique. For example, when discussing the relationship between alcohol intake and fatigue, the authors only refer to Woolley et al. (2004) and Vella et al. (2010). Furthermore, these articles seem to be off the topic. Woolley et al. examine whether chronic fatigue syndrome patients reduce or cease alcohol intake, and Vella et al. examine how alcohol consumption is related to exercise performance and recovery. In this paper, the authors examine whether alcohol consumption causes fatigue. It was also unclear how the study by Juster et al. (2010) was related to the authors' topic regarding smoking and fatigue.

4. There is evidence that alcohol consumption is related to poorer sleep. Could that also explain the positive link between alcohol consumption and fatigue?

5. In Figure 4, the authors report the Wald ratio for the “never drinking” outcome. In the table notes, the authors should inform the reader why this statistic was reported in this case.

6. How many SNPs were related to self-reported tiredness? This was not mentioned in the section "Data sources and processing".

7. Some of the abbreviations were not explained in the text.

Reviewer #2: Thank you for the opportunity to review the manuscript entitled, “Exploring causal effects of smoking and alcohol related lifestyle factors on self-report tiredness: a Mendelian randomization study,” by Li et al. In this study, the authors use Mendelian randomization methods to assess the impact of the genetic liability for tobacco smoking and alcohol consumption on self-reported tiredness. It is generally well-written and has some interesting findings, especially for the substance use field. I do have some comments that I believe will help the manuscript.

My main comments relate to the need to account for psychiatric comorbidities, at least in some way, in the study. For example, there are clinical comorbidities and genetic relationships (i.e., correlations) between major depression and substance use, and I can see causal pathways where it major depression may impact your smoking/fatigue findings. This would have important implications for the interpretation of the results and possible use for public health strategies: should you target the psychiatric comorbidity or the substance use behavior?

As the smoking and alcohol instruments are quite strong and contain genetic variants located throughout the genome, they may have corresponding associations with psychiatric endpoints, which if not accounted for in the analyses, may impact the findings. I believe this study would be greatly enhanced by a test of the robustness of the alcohol and smoking findings with a multivariable MR analysis that would identify whether there are independent effects of smoking and alcohol on fatigue. Multivariable MR is quite straightforward at this point (https://mrcieu.github.io/TwoSampleMR/reference/mv_multiple.html) and including several models with prevalent and comorbid psychiatric disorders (e.g., major depression, bipolar disorder, Schizophrenia, etc.) to test whether smoking and alcohol liability remain statistically significant would be valuable. I would like to see this added, at least as a sensitivity analysis, to the manuscript.

I have worked with the alcohol intake frequency data from the UK Biobank before and I believe it is reverse coded – higher categories correspond to reduced alcohol intake frequency (https://biobank.ctsu.ox.ac.uk/crystal/coding.cgi?id=100402). As the authors report a finding with alcohol intake frequency, can they please double check that the direction is correct?

6. PLOS authors have the option to publish the peer review history of their article (what does this mean?). If published, this will include your full peer review and any attached files.

Reviewer #1: No

Reviewer #2: No

---

## [Author Response · Author response to Decision Letter 0]

1 Apr 2023

Response to reviewer

Reviewer #1:

1. Plos One is an interdisciplinary journal and thus not all readers might be familiar with the MR method. Therefore, the idea of the two-sample MR and MR, in general, should be explained. 

Please refer to the corresponding part of revised manuscript (method section on pages 3 and 4.).

2. In the two-sample MR, the evidence on the associations of genetic variants with the risk factor and the outcome should come from nonoverlapping data sources. It seems that in this study, both associations come from the same sample (UKBB). Don’t the samples overlap? If so, what are the consequences? 

Please refer to the corresponding part of revised manuscript (Strengths and limitations section on page 15).

3. (a) There are very few citations to the previous literature regarding the relationship between smoking, alcohol intake and fatigue. For example, when discussing the relationship between alcohol intake and fatigue, the authors only refer to Woolley et al. (2004) and Vella et al. (2010). 

(b) Furthermore, these articles seem to be off the topic. Woolley et al. examine whether chronic fatigue syndrome patients reduce or cease alcohol intake, and Vella et al. examine how alcohol consumption is related to exercise performance and recovery. In this paper, the authors examine whether alcohol consumption causes fatigue. 

(c) It was also unclear how the study by Juster et al. (2010) was related to the authors' topic regarding smoking and fatigue. 

In the following parts, I will give you the explanation corresponding to the comments above step by step.

(a) As you mentioned, there are very few citations to the previous literature. There are indeed very few literatures on the topic. We have investigated the subject via searching on the PubMed and Google scholar websites, and indeed found very few literatures on such topic.

(b) I found the topic of Woolley et al. is related to mine, if you don’t mind me saying so. In the abstract of their article, Woolley et al. gave the reason for reducing alcohol intake [The most common reasons were increased tiredness after drinking (67%), increased nausea (33%), exacerbated hangovers (23%) and sleep disturbance (24%)]. Their findings support my research on the basis of the association between fatigue and alcohol intake.

Vella et al. reviewed how alcohol consumption is related to exercise performance (I). The exercise performance includes endurance performance (II), which is a core features to define physical fatigue (III), i.e. the inability to continue an endurance activity. My study is to investigate whether there is the causal relationship between alcohol consumption and (III) which is directly related to (II) and partly related to (I).

(c) I explained how the study by Juster et al. was related to my topic on page 12 based on the content of Juster et al. (2010). In section 4.2, Juster et al. gave us the following two findings: The first one is that smoking interacts with greater allostatic load. Another one is that increased fatigue is related to higher allostatic load. 

4. There is evidence that alcohol consumption is related to poorer sleep. Could that also explain the positive link between alcohol consumption and fatigue?

No. The reason is that alcohol consumption may have negative horizontal pleiotropy related to fatigue via other mediators rather than sleep.

5. In Figure 4, the authors report the Wald ratio for the “never drinking” outcome. In the table notes, the authors should inform the reader why this statistic was reported in this case.

See Figure 4 note in revised manuscript on page 11.

6. How many SNPs were related to self-reported tiredness? This was not mentioned in the section "Data sources and processing". 

See “Data sources and processing” section in revised manuscript on page 6.

7. Some of the abbreviations were not explained in the text.

 We examined the abbreviations in the manuscript carefully and revised. 

Reviewer #2:

1. I am very appreciated for your valuable suggestions. Multivariable Mendelian Randomization was performed as additional sensitivity analyses to test the robustness of the findings from univariate MR in method section on page 8. Other sections were revised correspondingly on pages 6, 10, 11. 

2. I am very sorry to tell you I have no other clues to check the direction for the alcohol intake frequency data in the UK Biobank. But the finding, i.e. the causal effect of alcohol intake frequency and alcohol drinking states on fatigue, might provide some evidence to support your guess – higher categories correspond to reduced alcohol intake frequency.

---

## [Decision Letter · Decision Letter 1]

17 Apr 2023

PONE-D-22-28356R1Exploring causal effects of smoking and alcohol related lifestyle factors on self-report tiredness: a Mendelian randomization studyPLOS ONE

Dear Dr. Song,

Thank you for submitting your manuscript to PLOS ONE. After careful consideration, we feel that it has merit but does not fully meet PLOS ONE’s publication criteria as it currently stands. Therefore, we invite you to submit a revised version of the manuscript that addresses the points raised during the review process.

The revised version should address all remaining concerns.

We look forward to receiving your revised manuscript.

Kind regards,

Petri Böckerman

Academic Editor

PLOS ONE

**Additional Editor Comments:**

The revised version should address all remaining concerns.

Reviewers' comments:

Reviewer's Responses to Questions

**Comments to the Author**

1. If the authors have adequately addressed your comments raised in a previous round of review and you feel that this manuscript is now acceptable for publication, you may indicate that here to bypass the “Comments to the Author” section, enter your conflict of interest statement in the “Confidential to Editor” section, and submit your "Accept" recommendation.

Reviewer #1: (No Response)

Reviewer #2: All comments have been addressed

2. Is the manuscript technically sound, and do the data support the conclusions?

Reviewer #1: Partly

Reviewer #2: Yes

3. Has the statistical analysis been performed appropriately and rigorously? 

Reviewer #1: I Don't Know

Reviewer #2: Yes

4. Have the authors made all data underlying the findings in their manuscript fully available?

Reviewer #1: Yes

Reviewer #2: Yes

5. Is the manuscript presented in an intelligible fashion and written in standard English?

Reviewer #1: Yes

Reviewer #2: Yes

6. Review Comments to the Author

Reviewer #1: Referee report on "Exploring causal effects of smoking and alcohol related lifestyle factors on self-report tiredness: a Mendelian randomization study" (PONE-D-22-28356).

I have the following comments on the revised manuscript.

1. Out of the blue, at the beginning of the Data Sources and Processing section (page 5, line 100), the authors start discussing psychiatric disorders. The rationale for using these variables should have been given much earlier. What is the purpose of using these variables?

2. The interpretation of the Multivariable Mendelian Randomization (MVMR) results needs clarification. Could you please explain why MVMR was performed? Was it to investigate whether the link between SAIEs and fatigue was mediated by psychiatric disorders? Are the reported estimates the remaining direct effects after controlling for the mediated pathway via psychiatric disorders? If so, why do the authors indicate on page 10 (lines 211-212) that the MVMR results did not support the idea of a causal relationship between current smoking and fatigue? It should be noted that a mediated effect via psychiatric disorders is still part of the causal effect. Even if there is no direct effect after controlling for a mediated effect, a significant total effect may still exist. It seems to me that MVMR was performed to provide an explanation for the relationship between SAIEs and fatigue.

3. Regarding the previous comments, could you please clarify the meaning of the following sentence (page 10, lines 212-215): The discrepancy with our results – suggesting mediation or confounding by psychiatric disorders – highlights the importance od the inclusion for all the SNPs significantly associated with substance use and psychiatric disorders in multivariable MR analysis, to prevent loss of precision and potentially even biased estimates.”

4. The MVMR results were added to the methods and results sections without being fully integrated into the rest of the paper. The authors should revise the paper to better incorporate the MVMR results and their implications into the overall discussion and conclusions.

5. Page 7, line 140: “to avoid the violation of the assumptions 2 and 3, we assessed instrument strength…” Instrument strength is related to assumption 1. Could you please clarify why assumptions 2 and 3 are mentioned here?

6. On page 7, line 144, the statement 'To test whether all the IVs satisfy the IV assumptions…' needs clarification. Could you please specify which assumptions you are referring to?"

7. Fig 1: It appears that IV assumption 2 and 3 should be the other way around.

8. Page 6 line 126: should the p-value be p < 0.05 instead of p > 0.05?

9. Pages 8-9 lines 181-182. The authors referred to Corwin et al. (2002) as a reference for the multivariable MR method assumptions, but this paper is not relevant to the multivariable MR method. Could you please provide a better reference for the multivariable MR method assumptions?

10. The authors reported the results for never-smoking status first, followed by current smoking status in the text, but the order of the results is reversed in Figure 2 (and a similar situation in Figure 3). To maintain consistency, the results should be presented in the same order in both the text and tables.

11. Page 10 line 212: I believe that “multivariable IV MR” should be “IVW multivariable MR analysis” for consistency.

12. Figure 4: There appears to be a discrepancy between the information in the legend and the footnote. According to the legend, the orange results indicate the MVMR results, but the footnote indicates that the blue results report the results adjusted for psychiatric disorders.

13. The explanation for the abbreviation PD is missing in the notes of Fig 5.

14. The phrase "we found strong evidence" should be toned down as the MR method has its limitations, which may not have been fully addressed in this study.

15. In the limitations section, the authors should acknowledge that the presence of pleiotropy as a potential factor affecting their results cannot be ruled out.

References

Corwin, E. J., Klein, L. C., & Rickelman, K. (2002). Predictors of fatigue in healthy young adults: moderating effects of cigarette smoking and gender. Biological research for nursing, 3(4), 222-233.

Reviewer #2: The authors have responded to my original concerns/comments. The revised manuscript is much improved.

7. PLOS authors have the option to publish the peer review history of their article (what does this mean?). If published, this will include your full peer review and any attached files.

Reviewer #1: No

Reviewer #2: No

---

## [Author Response · Author response to Decision Letter 1]

16 May 2023

Response to reviewer

Reviewer #1:

 I am very appreciated for your serious and careful review as this is my first paper in bioinformatics. With your help, I will achieve a satisfied paper, and Mendelian Randomization will become a flexible tool for me. I believe I have made much progress. Thank you again for your time! 

1. Out of the blue, at the beginning of the Data Sources and Processing section (page 5, line 100), the authors start discussing psychiatric disorders. The rationale for using these variables should have been given much earlier. What is the purpose of using these variables?

Please refer to the corresponding part of revised manuscript (introduction section on page 3, line 68-71.).

2. The interpretation of the Multivariable Mendelian Randomization (MVMR) results needs clarification. 

（a） Could you please explain why MVMR was performed?

Please refer to the corresponding part of revised manuscript (introduction section on page 3, line 68-71.). I explained the reason for performed MVMR.

（b） Was it to investigate whether the link between SAIEs and fatigue was mediated by psychiatric disorders? 

No, this paper is to investigate whether there is a direct link between SAIEs and fatigue not via psychiatric disorders. In fact, I intend to exclude the effect of mental illness on the direct casual estimate between SAIEs and fatigue. I apply the MVMR method which excludes psychiatric disorders as confounders and mediators.

（c） Are the reported estimates the remaining direct effects after controlling for the mediated pathway via psychiatric disorders? 

Yes, as you mentioned, MVMR results show the remaining direct effects after controlling for the mediated pathway via psychiatric disorders. Besides this, the method MVMR also eliminated the possibility of psychiatric disorders to be confounders for SAIEs and fatigue.

（d） If so, why do the authors indicate on page 10 (lines 211-212) that the MVMR results did not support the idea of a causal relationship between current smoking and fatigue? 

For the exposure (current smoking: status), the confidence interval for the odds ratio obtained from MVMR is including “1”, i.e., “1” represents no direct causal relationship; on the contrary, the same parameter from univariate MR is not including “1”.

（e） It should be noted that a mediated effect via psychiatric disorders is still part of the causal effect. Even if there is no direct effect after controlling for a mediated effect, a significant total effect may still exist. It seems to me that MVMR was performed to provide an explanation for the relationship between SAIEs and fatigue.

As you mentioned, if there is no overlapping between the instrument variables (SNPs) for exposures, MVMR findings can provide an explanation for the relationship between SAIEs and fatigue. But the instrument variables of SAIEs are quite strong and contain genetic variants located throughout the genome, they may have corresponding associations with psychiatric endpoints. Therefore, generally there is overlapping SNPs between SAIEs and psychiatric disorders. Such overlapping SNPs lead to false positives causal estimate of univariate MR.

3. Regarding the previous comments, could you please clarify the meaning of the following sentence (page 10, lines 212-215): The discrepancy with our results – suggesting mediation or confounding by psychiatric disorders – highlights the importance of the inclusion for all the SNPs significantly associated with substance use and psychiatric disorders in multivariable MR analysis, to prevent loss of precision and potentially even biased estimates.”

The discrepancy with our results, i.e., there are different results between MVMR and univariate MR for the exposure (current smoking: status). 

suggesting mediation or confounding by psychiatric disorders gives the reason for the discrepancy with our results (MR and MVMR). More clearly the discrepancy is due to the two basic relationships (confounder and meditator) between psychiatric disorders and SAIEs.

the inclusion for all the SNPs significantly associated with substance use and psychiatric disorders in multivariable MR analysis, i.e., one assumption of MVMR (exposures must be strongly predicted by the SNPs “given the other exposures included in the model”). This assumption ensures MVMR to eliminate possible bias from mediation or confounding by psychiatric disorders in univariate MR. 

4. The MVMR results were added to the methods and results sections without being fully integrated into the rest of the paper. The authors should revise the paper to better incorporate the MVMR results and their implications into the overall discussion and conclusions. 

I added corresponding revision in the discussion section on page 12 (line 257-258, 261-262,265-271), page 13 (291-292), page 14 (line 293-294, line 307-308), page 15 (line 329); and in conclusion section on page 16, line 338.

5. Page 7, line 140: “to avoid the violation of the assumptions 2 and 3, we assessed instrument strength…” Instrument strength is related to assumption 1. Could you please clarify why assumptions 2 and 3 are mentioned here?

The effect of instrument strength on MR test is not only related to assumption 1. Weak instruments that poorly predict the risk factor can amplify the bias due to violations of the core instrumental variable assumptions, i.e., horizontally pleiotropic effects of variants and confounders. 

6. On page 7, line 144, the statement 'To test whether all the IVs satisfy the IV assumptions…' needs clarification. Could you please specify which assumptions you are referring to?"

Please refer to the revised corresponding part: Page 7, line 148.

7. Fig 1: It appears that IV assumption 2 and 3 should be the other way around.

Sorry, I am not sure if I understand this comment fully. In fact, there is no fixed order for assumption 2 and 3. I checked my paper carefully and searched figures relevant to MR through internet. Most figures about assumption 2 and 3 are in the same order of mine. 

8. Page 6 line 126: should the p-value be p < 0.05 instead of p > 0.05?

Yes, the p-value should be p < 0.05. Thanks a lot!

9. Pages 8-9 lines 181-182. The authors referred to Corwin et al. (2002) as a reference for the multivariable MR method assumptions, but this paper is not relevant to the multivariable MR method. Could you please provide a better reference for the multivariable MR method assumptions?

I am very sorry for this typesetting error. In fact, the order of Corwin et al. (2002) in the reference list is 31 in the paper revised last time. Another reference no.30 is the corresponding one for MVMR method in that paper.

10. The authors reported the results for never-smoking status first, followed by current smoking status in the text, but the order of the results is reversed in Figure 2 (and a similar situation in Figure 3). To maintain consistency, the results should be presented in the same order in both the text and tables.

Please to refer the revised Figure 2 and 3.

11. Page 10 line 212: I believe that “multivariable IV MR” should be “IVW multivariable MR analysis” for consistency. 

Thanks for your help! Please refer to the revision on page 10 (line 216).

12. Figure 4: There appears to be a discrepancy between the information in the legend and the footnote. According to the legend, the orange results indicate the MVMR results, but the footnote indicates that the blue results report the results adjusted for psychiatric disorders.

Thanks for your help, please refer to the revision on page 11 (line 232).

13. The explanation for the abbreviation PD is missing in the notes of Fig 5. 

Thanks for your help! Please refer to the revision on page 11 (line 235).

14. The phrase "we found strong evidence" should be toned down as the MR method has its limitations, which may not have been fully addressed in this study. 

Please refer to the revision on page 12 (line 257). Thank you !

15. In the limitations section, the authors should acknowledge that the presence of pleiotropy as a potential factor affecting their results cannot be ruled out.

please refer to the revision on page 15 (line 329). Thank you!

---

## [Decision Letter · Decision Letter 2]

29 May 2023

Exploring causal effects of smoking and alcohol related lifestyle factors on self-report tiredness: a Mendelian randomization study

PONE-D-22-28356R2

Dear Dr. Song,

We’re pleased to inform you that your manuscript has been judged scientifically suitable for publication and will be formally accepted for publication once it meets all outstanding technical requirements.

Kind regards,

Petri Böckerman

Academic Editor

PLOS ONE

Additional Editor Comments (optional):

I am happy with the revised version of the paper.

Reviewers' comments:

Reviewer's Responses to Questions

**Comments to the Author**

1. If the authors have adequately addressed your comments raised in a previous round of review and you feel that this manuscript is now acceptable for publication, you may indicate that here to bypass the “Comments to the Author” section, enter your conflict of interest statement in the “Confidential to Editor” section, and submit your "Accept" recommendation.

Reviewer #1: (No Response)

2. Is the manuscript technically sound, and do the data support the conclusions?

Reviewer #1: (No Response)

3. Has the statistical analysis been performed appropriately and rigorously? 

Reviewer #1: (No Response)

4. Have the authors made all data underlying the findings in their manuscript fully available?

Reviewer #1: (No Response)

5. Is the manuscript presented in an intelligible fashion and written in standard English?

Reviewer #1: (No Response)

6. Review Comments to the Author

Reviewer #1: (No Response)

7. PLOS authors have the option to publish the peer review history of their article (what does this mean?). If published, this will include your full peer review and any attached files.

Reviewer #1: No

---

## [Editor Report · Acceptance letter]

6 Jun 2023

PONE-D-22-28356R2 

Exploring causal effects of smoking and alcohol related lifestyle factors on self-report tiredness: a Mendelian randomization study 

Dear Dr. Song:

I'm pleased to inform you that your manuscript has been deemed suitable for publication in PLOS ONE. Congratulations! Your manuscript is now with our production department. 

Kind regards, 

on behalf of

Professor Petri Böckerman 

Academic Editor

PLOS ONE